# Microalgal colony blot: A simple and rapid method for direct detection of recombinant protein production in microalgae colonies

**Yasin Torres-Tiji**[ORCID]¤*, **Francis J. Fields, Stephen P. Mayfield**

School of Biological Sciences, University of California San Diego, La Jolla, California, United States of America

¤ Department of Chemistry, Colorado School of Mines, Golden, United States of America
* yasin.torres-tiji@mines.edu

## Abstract

Microalgae have emerged as a versatile biotechnological platform, promising to become an efficient source of commodities such as food, feed, and biofuels, but these organisms have also sparked profound scientific and commercial interest for their potential in producing high-value recombinant proteins. Recombinant gene expression is highly dependent on the loci in which the transgene integrates, and in green algae transgenes integrate randomly into the nuclear genome, mostly through non homologous end joining. Therefore, recombinant gene expression varies greatly among different algal transformants and many algal colonies must be screened before a suitable production strain can be found, which can be quite laborious and become a bottleneck in the recombinant strain production pipeline. Here we describe a method for mid-throughput screening of recombinant protein expression; the Microalgal Colony Blot. This screening method allows for the detection of recombinant protein expression in up to 100 algal colonies per petri dish, with each petri dish preparation taking only 20 minutes. A nitrocellulose membrane is layered on top of a petri dish containing agar media, and algal cells are inoculated on top of the filter in a liquid suspension using a micropipette. The colonies are allowed to grow for up to 7 days, with the colonies secreting recombinant protein (either through active secretion or through cell lysis) as they grow with the recombinant protein being immediately bound by the nitrocellulose membrane. After the incubation period, the membrane is treated like a regular western blot, with blocking, washing, antibody binding and visualization. In this manner, up to 1000 colonies can be comfortably screened per day by a single person. Knowing that in *C. reinhardtii* only about 5% of the transgenic colonies from a transformation produce significant recombinant protein expression, being able to screen 1000 colonies ensures that around 50 suitable candidates will be identified within a single day.

**Data availability statement:** All relevant data are within the manuscript and its Supporting Information files, with additional supporting datasets available on Zenodo under a Creative Commons Attribution 4.0 International license (CC BY 4.0). These datasets include plasmid sequence files, flow cytometry data, and uncropped original blot images and are available at: https://doi.org/10.5281/zenodo.18439538.

**Funding:** The funding that made this work possible was granted by the US Department of Energy – Bioenergy Technologies Office under grant number: DE-EE0009671 (APEX). The funder provided support in the form of salaries for authors FF, YTT, and SM. The funders had no role in study design, data collection and analysis, decision to publish, or preparation of the manuscript. The specific roles of these authors are articulated in the 'author contributions' section.

**Competing interests:** Stephen Mayfield is CEO of, board member of, and holds equity in Algenesis Inc., a company developing microalgae-based products. Francis Fields is a contractor for the US Department of Energy, which administers funding for algae-related research projects. The authors declare no patents, products in development, or marketed products directly related to this methodology. This does not alter our adherence to PLOS ONE policies on sharing data and materials.

## Introduction

Microalgae have been used as a host for the biotechnological production of varied biological goods like food additives (food pigments, gelling agents, etc.), nutraceuticals (pro-vitamin A, omega-3 fatty acids, β-carotene, etc.), cosmetics, animal feed, etc [1]. Additionally, microalgae have incredible potential at synthesizing recombinant products like proteins, or metabolites produced through metabolic engineering [2]. The model organism *Chlamydomonas reinhardtii* has been used to successfully produce multiple recombinant proteins, engineering the chloroplast genome or the nuclear genome [3]. Most research in this area focused on the usage of the chloroplast genome due to its ability to integrate transgenes through homologous recombination [4]. Since transgene expression heavily depends on the locus of integration, the ability to target transgenes to a desired locus benefits recombinant protein expression in the chloroplast. However, the chloroplast is not able to add most post-translational modifications like glycosylation, and proteins are not able to leave the organelle which represents an obstacle for metabolic engineering of biochemical pathways outside the chloroplast [3]. Therefore, research has turned to engineering the nuclear genome to harbor transgenes.

In the nuclear genome of *C. reinhardtii*, transgenes integrate randomly through Non-Homologous End Joining. Consequently, transgene expression varies greatly among different transformants depending on the loci of integration, and hundreds of colonies need to be screened for recombinant protein expression [5]. An effective solution to this problem has been the utilization of Fluorescence Activated Cell Sorting, which allows for the screening of thousands of cells per second. This requires the use of a fluorescent protein, which can be used on its own as a reporter gene or can be fused to a protein of interest [6]. However, sometimes the needs of the recombinant protein of interest cannot accommodate its fusion to a fluorescent reporter. High rates of transgene truncation in *C. reinhardtii*, combined with strong selection pressures favoring only part of the transgene, can lead to the selection of transformants with high fluorescent expression that possess truncated transgenes, resulting in the expression of only the fluorescent segment. Therefore, if the recombinant protein of interest is not fluorescent nor chemiluminescent, the screening needs to be done through a direct detection method, such as western blot, which can be tedious and become a bottleneck in the production of algal recombinant strains.

Dot-blot assays originated as a streamlined alternative to Southern and Northern blotting for detecting DNA and RNA sequence homologies. Rather than first separating nucleic acids by gel electrophoresis, samples were simply "spotted", in the form of a dot, directly onto nitrocellulose membranes and hybridized with labeled DNA or RNA probes [7]. This same approach was soon adapted to proteins, using specific antibodies as molecular recognition elements to capture and visualize antigens in an immunoassay format [8,9]. The method was later adapted for recombinant protein detection from *Escherichia coli* clones by directly transferring and lysing colonies on nitrocellulose, then blotting the liberated proteins onto CNBr-activated paper for subsequent detection [10]. Over time, these methods

were refined to detect an ever-wider range of targets and, with the appropriate modifications, to yield semi-quantitative or fully quantitative results [11].

Here we present a variation of the Colony Blot [10], the "Microalgal Colony Blot" (MCB). In this approach, *C. reinhardtii* is transformed with a recombinant protein expression vector, and the resulting colonies are patched onto a gridded agar plate and allowed to grow for 3–5 days approximately. A nitrocellulose membrane, with a pencil drawn grid resembling the grid of the patched agar plate, is placed atop a fresh agar plate so that it covers the entirety of the agar surface. Once the patched colonies have accumulated enough biomass, the algal biomass is carefully placed on top of the nitrocellulose membrane in the same grid order as the plate of origin. The colonies are incubated on the membrane for 1–7 days. During this period, the cells secrete the recombinant protein, which is immediately bound by the membrane.

The technique was originally designed to detect colonies that secreted recombinant protein, because assaying protein diluted in liquid-culture supernatants was very time-consuming as it required protein precipitation to enrich total protein content [12,13]. However, we later discovered that it also detects intracellular protein accumulation: as the cells grow on top of the nitrocellulose membrane, a fraction die and spill their intracellular contents onto the membrane. So it is within this concept that the crucial aspect of the technique lies: dynamic incubation time. Longer periods for low yields of recombinant protein expression and shorter periods for high yields, analogous to a camera's exposure setting. By depositing an equivalent patch of algal colony onto the membrane, each underlying unit area becomes uniformly saturated with cells at the same density, ensuring that every dot delivers the same number of cells per unit membrane area and thus enabling reliable, semi-quantitative comparison of signal intensities. After incubation, the cells are rinsed from the membrane with care to remove cell debris without damaging the nitrocellulose, and then the membrane is processed like a standard western blot following the protein-transfer step.

The MCB thus provides a rapid, straightforward, and robust approach for the direct, semi-quantitative detection of recombinant proteins in *C. reinhardtii*, and potentially other microalgae. By enabling medium-throughput screening of hundreds to thousands of colonies in a matter of days, it effectively overcomes the primary bottleneck of identifying high-expressing algal strains.

## Methods

### Algal strain, media and growth conditions

The algal species used in this study, unless otherwise specified, was *Chlamydomonas reinhardtii*, strain cc-1690. Other algal species tested in Fig 5 are *C. reinhardtii* strain CR25 [12] and *Chlamydomonas pacifica* strain 402 [14]. The algae were cultured in Erlenmeyer flasks containing Tris-Acetate Phosphate (TAP) media under continuous illumination with a photosynthetically active radiation intensity of 125 µE/m²/s. The algal cultures were constantly agitated at 125 RPM on an orbital shaker table at 25°C.

### Plasmids and Transformation

Several plasmids were used in this experiment. For the main part of the manuscript (Figs 1-4), two plasmids were used: a test plasmid and a control plasmid. The test plasmid expresses mClover GFP, serving as the reporter gene to evaluate the MCB procedure described herein. Its expression is driven by the AR1 promoter and the 5' UTR of the rbcs2 gene from *C. reinhardtii*. The coding sequence (CDS) comprises a fusion of a codon-optimized Ble gene, which confers zeocin resistance, and mCloverGFP, with a Flag® tag appended. Notably, the Ble gene includes the first intron of rbcs2 inserted mid-gene, and the mCloverGFP sequence incorporates the second intron of rbcs2; these intron insertions have been shown to enhance recombinant expression in *C. reinhardtii*. The CDS is followed by the rbcs2 3' UTR and is accompanied by a second open reading frame encoding a Hygromycin resistance gene, although Hygromycin selection was not employed in this study. This plasmid was first described by Sproles et al. (2022) [6], and is illustrated in S2 Fig.

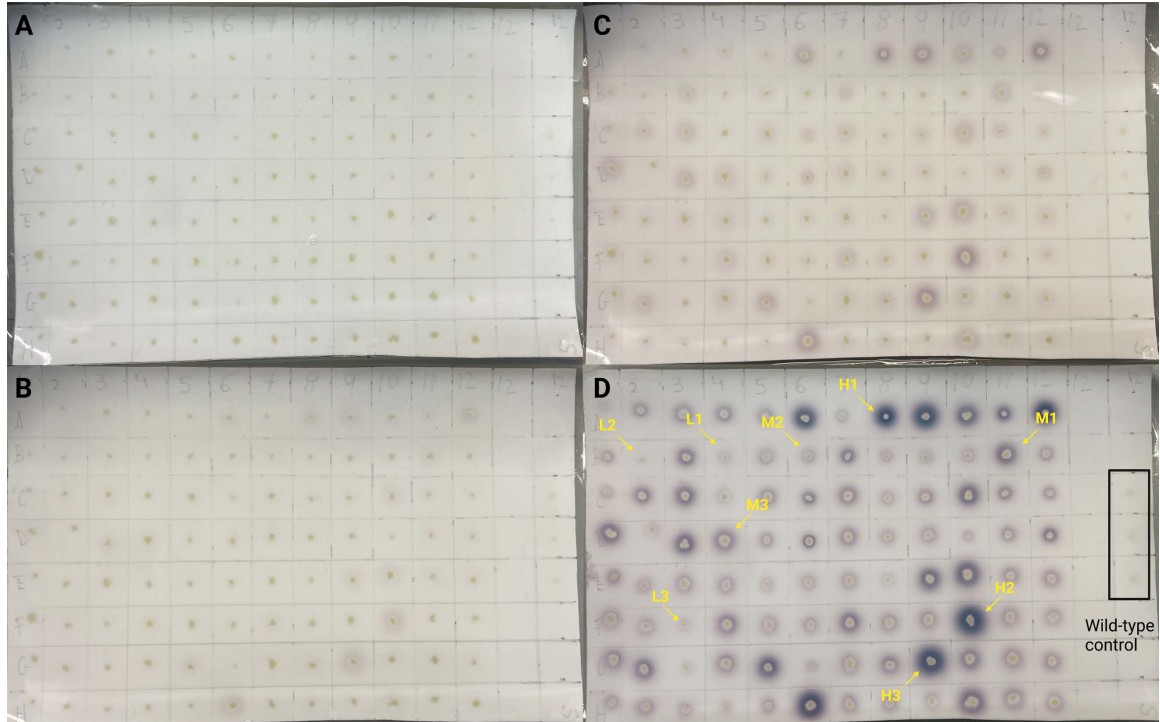

**Fig 1. Microalgal Colony Blot.** The blot was incubated with an alkaline phosphatase-conjugated anti-GFP antibody and stained with NBT/BCIP. Panel A shows the membrane immediately after NBT/BCIP addition; Panel B at 1 min; Panel C at 3 min; and Panel D at 15 min. Three wild-type control colonies (far right) exhibit minimal staining. The remaining colonies display a gradient of signal intensity, from which three high (H1-H3), three medium (M1-M3), and three low (L1-L3) signal intensity colonies were selected for further analysis.

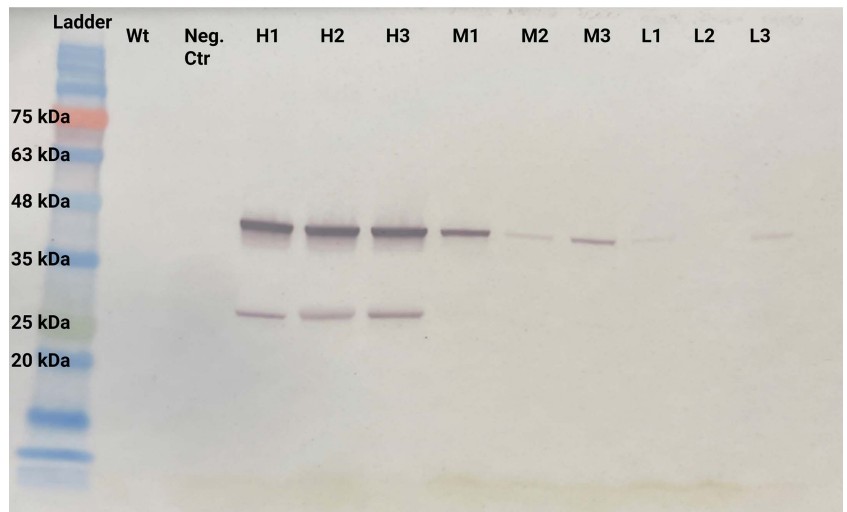

**Fig 2. Western Blot of selected transformants.** The membrane was probed with the same anti-GFP alkaline phosphatase-conjugated antibody used for the colony blot. Each lane was loaded with 20 µg of total soluble protein. The Ble-GFP-Flag® fusion protein (40.5 kDa) corresponds to the upper band, while the lower 26.5 kDa band reflects cleavage between Ble and GFP-Flag®. Clones H1-H3 exhibit high levels of the GFP fusion, M1-M3 show intermediate levels, and L1-L3 display low levels. Lanes Wild-type and Negative Control show lack of any unspecific bands.

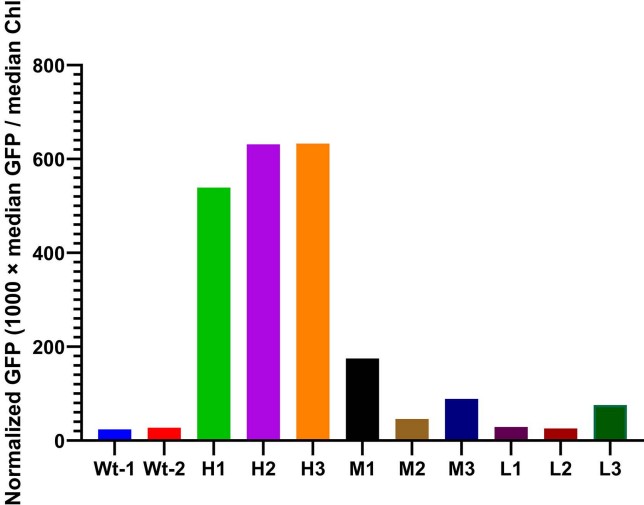

**Fig 3. Flow cytometry analysis of clonal populations derived from selected transformants.** For each population, 15,000-80,000 events were acquired and gated to exclude non-viable cells based on chlorophyll autofluorescence measured in the FL3 (Auto-APC-A) channel. Median GFP fluorescence values were normalized to median chlorophyll fluorescence to account for differences in cell size and aggregation, and the resulting ratios were multiplied by 1,000 for ease of interpretation.

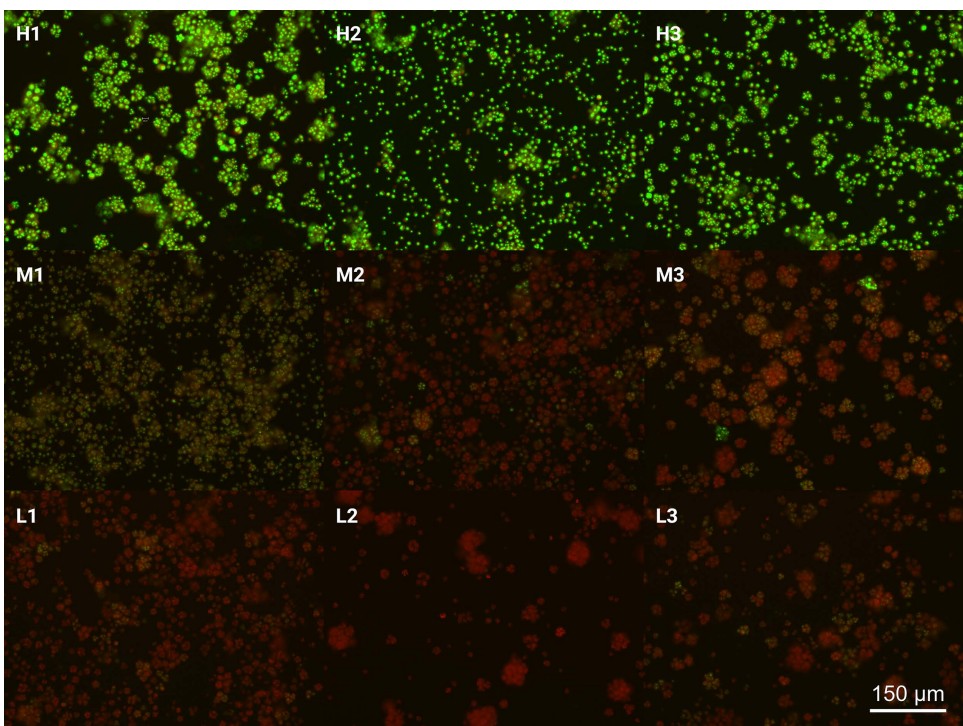

**Fig 4. GFP fluorescence in selected transformants.** The previously isolated clones were imaged at 200X magnification in an inverted fluorescent microscope (EVOS™ M5000, ThermoFisher Scientific). The images shown are the result of merging the GFP channel (colored green, showing GFP fluorescence) and the Texas Red channel (colored red, showing chlorophyll autofluorescence). Exposure and gain settings were identical across all images for direct comparison. Clones H1-H3 show a remarkably high GFP signal. Clones M1-M3 show distinguishable GFP signal, albeit much more reduced than H1-H3. Clones L1-L3 show minimal GFP signal.

The control plasmid expresses an algal codon-optimized, truncated human ICAM-1 protein transcriptionally fused to the Ble gene. Similar to the test plasmid, its CDS is controlled by the AR1 promoter and the 5' UTR of rbcs2. The Ble and ICAM-1 genes are separated by a Foot-and-Mouth Disease Virus 2A (FMDV2A) self-cleaving peptide. Additionally, a secretion peptide derived from the Aryl Sulfatase 1 (ARS1) gene of *C. reinhardtii* is fused to the N-terminus of the ICAM-1 protein, and a Flag® tag is positioned at the 5' end of the CDS, which concludes with the rbcs2 3' UTR. This plasmid was first described by Torres-Tiji et al. (2022) [13], and is illustrated in S2 Fig.

For Fig 5-A and B, the previously described control plasmid was utilized. In Fig 5-C and D, positive controls consisted of strains harboring plasmids expressing the SARS-CoV-2 receptor-binding domain (RBD; amino acids 319–537) fused to mClover; these were codon-optimized for *C. reinhardtii* and targeted for either secretion via a PHC2 signal peptide or ER-retention using a C-terminal KDEL motif [15]. The experimental strains harbored a plasmid in which the mature peptide of the Vibrio cholerae enterotoxin subunit B (CtxB; residues 22–124, UniProt P01556) was N-terminally fused to the RBD. In this configuration, the PHC2 signal peptide was maintained at the N-terminus for secretory pathway entry, followed by the CtxB sequence and the RBD (residues 319–537) at the C-terminus, creating a CtxB:RBD fusion protein within the same vector backbone used for the standalone RBD controls. These plasmids are illustrated in S2 Fig. Transformations were performed by electroporation following the protocol described in Molino et al. (2018) [16]. The plasmids

**Fig 5. Microalgal Colony Blot detection of alternative proteins and species.** Microalgal colony blots before (left panels) and after (right panels) antibody incubation and detection. **(A-B)** MCB in *C. reinhardtii* detecting expression and secretion of a truncated human ICAM1 protein, probed with an anti-Flag® Alkaline Phosphatase-conjugated antibody. **(C-D)** MCB in *C. reinhardtii* detecting expression and secretion of a CtxB::RBD fusion protein, probed with a primary anti-RBD antibody and a secondary anti-rabbit Alkaline Phosphatase antibody. The bottom row shows wild-type C. reinhardtii spotted in triplicate, and the second-to-last row shows positive controls in duplicate expressing an RBD::mCloverGFP fusion retained in the ER (first duplicate) or secreted (second duplicate). **(E-F)** MCB of *C. reinhardtii* and *C. pacifica* clones expressing mCloverGFP. The first square contains wild-type *C. reinhardtii* in triplicate, the second square contains a C. reinhardtii mCloverGFP-expressing positive control in triplicate, and the third square contains a *C. pacifica* mCloverGFP-expressing positive control in triplicate. The remaining squares contain individual *C. reinhardtii* clones expressing mCloverGFP.

were first digested with KpnI and XbaI, then purified using a Wizard® SV Gel and PCR Clean-Up System (Cat. #A9281, Promega, Madison, WI) and eluted in ultrapure DNase/RNase-free $H_2O$ to prevent salt interference during electroporation. The electroporated cells were then plated on TAP-agar plates containing 20 mg/L of Zeocin, and colonies were allowed to grow for a week.

## Microalgal colony blot

The protocol described in this peer-reviewed article is published on protocols.io, dx.doi.org/10.17504/protocols.io.kxygx9k-2kg8j/v1, and is included for printing as supporting information file 1 with this article.

Briefly, a nitrocellulose membrane marked with a corresponding pencil-drawn grid was placed atop a fresh Zeocin-containing TAP-agar plate, and 1–2 µL of biomass from each transformant was spotted onto the corresponding grid position. Nitrocellulose membranes were used, as preliminary tests with PVDF showed unacceptable background signal. As a wild-type negative control, Zeocin-resistant *C. reinhardtii* cells expressing a recombinant protein lacking cross-reactivity with the anti-GFP antibody were included. Plates were incubated at 30°C under constant light for 72 hours. Following incubation, membranes were washed with Tris-Buffered Saline with 0.1% Tween-20 (TBST), blocked with TBST containing 5% non-fat milk powder for 1 hour, and incubated with alkaline phosphatase-conjugated anti-GFP antibody (Cat. #ab6661, Abcam, Cambridge, UK; 1:2000 dilution) for 1 hour at room temperature. After washing, membranes were developed using NBT/BCIP substrate (Cat. #11697471001, Roche). The MCB seen in Fig 5-B was incubated using a monoclonal anti-Flag® Alkaline Phosphatase antibody (Cat. # A9469, Sigma-Aldrich, St. Louis, MO, USA) conjugated at a 1:3000 dilution in TBST containing 5% non-fat milk powder for 1 hour [13]. The MCB in Fig 5-D was incubated using a primary rabbit polyclonal anti-SARS-CoV-2-RBD antibody (Cat. #40592-T62 Sino Biological, Chesterbrook, PA) at a 1:3000 dilution in TBST+5% non-fat milk powder for 1 hour. After washing, the membrane was incubated with a goat anti-rabbit Alkaline Phosphatase conjugated secondary antibody (Cat. #31340; Thermo Fisher Scientific; Waltham, MA, USA) at a 1:10,000 dilution.

## Western Blot

Western blotting was performed as described in Torres-Tiji et al. (2022) [13], with minor modifications. Briefly, cell pellets were lysed by thoroughly resuspending them in lysis buffer at a 1:5 (pellet:buffer) volume ratio. The lysis buffer consisted of BugBuster 10 × Protein Extraction Reagent (Cat. #70584, MilliporeSigma, Burlington, MA) diluted 10-fold in 1 × Tris-Buffered Saline (TBS). The lysate was then centrifuged, and the supernatant containing the protein extract was collected. A Pierce™ BCA Protein Assay (Cat. #23225, ThermoFisher Scientific, Waltham, MA) was performed, following the manufacturer's instructions, to determine the total protein content. An amount of supernatant containing 20 µg of total protein was mixed (3:1, sample:loading buffer) with 4 × Laemmli buffer (Cat. #1610747, Bio-Rad Laboratories Inc., Hercules, CA) containing β-mercaptoethanol (9:1, Laemmli:β-mercaptoethanol). Samples were incubated at 70 °C for 10 min and loaded onto a 12% Mini-PROTEAN TGX precast gel (Cat. #4561035, Bio-Rad, Hercules, CA, USA), or 4–20% (Cat. #4561094, Bio-Rad, Hercules, CA, USA) for the western blot shown in S3 Fig, for SDS-PAGE at 200 V for 30 min. In the first well of each gel, 7 µL of AccuRuler RGB Plus Prestained Protein Ladder (Cat. # PM-001; Biopioneer, San Diego, CA, USA) was loaded. Proteins were then transferred onto a nitrocellulose membrane using a Mini Trans-Blot Cell (Cat. #1703930, Bio-Rad) at 200 mA for 1 h. Membranes were blocked, incubated with an Alkaline Phosphatase conjugated anti-GFP antibody (Cat. #ab6661, Abcam, Cambridge, UK), washed thoroughly, and developed using an NBT/BCIP substrate (Cat. #11697471001, Roche, Penzberg, Germany) according to the manufacturer's instructions. The western blot for detecting ICAM1 was incubated with the anti-Flag® antibody as previously described. The western blot for detecting the SARS-CoV-2-RBD protein was incubated with the primary and secondary antibodies as previously described. The western blot for detection of the CtxB protein was incubated with a rabbit polyclonal anti-cholera toxin B subunit antibody (Cat. #PA1–85293, Invitrogen, Thermo Fisher Scientific; Waltham, MA, USA) at a 1:1000 dilution, followed by incubation with the previously described goat anti-rabbit IgG alkaline phosphatase-conjugated secondary antibody.

## Flow cytometry

Flow cytometry was performed using a CytoFLEX cytometer (Beckman Coulter, Brea, CA, USA) with CytExpert 2.4 software. GFP fluorescence was measured using the FL1 channel (GFP FITC; 488 nm excitation; 525/40 nm emission filter) with a gain of 60. Chlorophyll autofluorescence was monitored using the FL3 channel (Auto APC-A; 638 nm excitation; 660/10 nm emission filter) with a gain of 600. Forward scatter (FSC) and side scatter (SSC) were acquired with gains of 500 and detection thresholds of 50,000 and 10,000, respectively. A minimum of 10,000 events were collected per sample. Events were first gated to include only viable cells, based on wild-type control samples (corresponding to a minimum APC-A autofluorescence threshold of approximately 770K). Within this viable population, the top 20% of GFP-positive events were selected for analysis (S1 Fig). Median GFP fluorescence was then normalized to median chlorophyll fluorescence to account for differences in cell size and cell aggregation, and the resulting values were multiplied by 1,000 for ease of interpretation.

## Inverted fluorescence imaging

Fluorescent images were acquired using an inverted microscope (EVOS™ M5000, Thermo Fisher Scientific, Waltham, MA) fitted with a 20 × objective lens (Evos, Part Number: AMEP4982). The objective lens used had a numerical aperture (NA) of 0.45 and a working distance of 6.12 mm, optimized for dry use, with color and flat-field correction enabled. No immersion oil was applied, as the lens was designed for dry imaging. The image acquisition was controlled using Evos2 software.

All images were captured with a resolution of 2048 × 1536 pixels, with a physical pixel size of 0.3085 µm/pixel. Exposure times were set to 50 ms for the GFP channel and 167 ms for the Texas Red channel. Gains were configured to be 30 for the GFP channel and 50 for the Texas Red channel. The photometric interpretation of the images was in the RGB color space, with 8 bits per sample for each channel, resulting in 24 bits per pixel. The physical size of the image was calibrated to a scale of 0.3085 µm per pixel.

Fluorescent channels were selected based on their emission spectra: the GFP channel was used to detect green fluorescence with an emission peak at 510 nm, and the Texas Red channel was employed to detect red fluorescence with an emission at 624 nm. Each channel was pseudo-colored according to standard laboratory protocols; the GFP channel was pseudo-colored in green (#ff00ff00) and the Texas Red channel in red (#ffff0000). Both channels were adjusted to a brightness of 50% and a contrast of 33%. The light source intensity was configured at 20% for the GFP channel and 17% for the Texas Red channel.

# Results

## Transformation and microalgal colony blot screening

*C. reinhardtii* was transformed with a plasmid expressing mCloverGFP fused to the Ble antibiotic resistance gene under control of the AR1 promoter. Following selection on Zeocin-containing media, colonies were patched and subjected to the MCB procedure. Pictures of the membrane while the colorimetric reaction took place were taken after 0, 1, 3 and 15 minutes (Fig 1).

As we can see in Fig 1, halos of indigo color, as a result of the alkaline phosphatase enzymatic reaction, appeared surrounding the colonies incubated on the nitrocellulose membrane. Different colonies present halos of different sizes and intensities. The size of the halo correlated to the size of the algal colony, but the intensity of the indigo color correlates with the amount of GFP blotted onto the membrane. Therefore, the colonies presenting halos with the highest intensities are the colonies that express the highest titers of GFP. Conversely, we can observe that the colonies for the negative control do not form a halo. Finally, we can see in Fig 1 that the highest expressors show visible halos after 1 minute, whereas longer incubation times are needed to visualize the halo of lower expressors.

To verify the correlation between halo indigo intensity and GFP expression, we selected three predicted high expressors (H1, H2 and H3), three predicted middle expressors (M1, M2 and M3), and three predicted low expressors (L1, L2 and L3), as can be seen labeled in Fig 1.

**Verification of Microalgal Colony Blot through western blot, flow cytometry and fluorescence imaging**

An important element for the successful detection of the protein of interest using MCB, just like in western blot, is that the antibody used has a strong affinity which will determine the sensitivity of the assay. However, unlike in western blot, there is no mass separation of proteins. Thus, high antibody specificity is crucial because unspecific binding will be undistinguishable to specific binding. This needs to be empirically determined through a western blot using the same antibody to be employed in the MCB, as well as the same washing, blocking and incubating conditions. For such verification, we performed a western blot on the soluble algal protein extract from a wild-type strain, the negative control strain used in the MCB, and the isolated strains H1, H2, H3, M1, M2, M3, L1, L2, and L3. Additionally, equal amounts of total soluble protein (20 μg) were loaded in each well to determine the relative abundance of Ble-GFP- Flag® fusion in each sample. As we can see in Fig 2, neither the wild-type strain nor the negative control strain showed any bands in the western blot, therefore ruling out unspecific binding. As for the rest of the bands, we can clearly see how strains H1, H2 and H3 show the highest levels of Ble-GFP-Flag®, detected as a 40.5 kDa band. In fact, so much GFP is detected that we also detect the cleaved GFP-Flag® product as a 26.5 kDa band. Next, we detect the Ble-GFP-Flag® band in the samples M1, M2 and M3 with a markedly lower intensity compared to the H1, H2 and H3 samples, albeit with significantly higher intensity than in the L1, L2 and L3 samples. And even though we can see some variation of Ble-GFP-Flag® intensity among the samples of medium expressors and among lower expressors, we can clearly see that on average the medium expressors show lower expression levels than the high expressors, but higher than the low expressors. Thus, western blot analysis confirmed the antibody's high specificity and revealed a consistent correlation between its signal intensities and those measured by the MCB.

For the detection and isolation of algal cells expressing fluorescent recombinant proteins, fluorescence-activated cell sorting (FACS) is widely regarded as the gold standard. To benchmark the performance of the MCB against flow cytometry, GFP fluorescence was measured in cell populations derived from clonal isolates classified as high (H1-H3), medium (M1-M3), and low (L1-L3) expressors, along with two independent wild-type cultures. As shown in Fig 3, wild-type replicates exhibited baseline normalized GFP fluorescence values of 24 and 27. In contrast, high-expressing clones displayed substantially elevated fluorescence, with normalized values of 539, 631, and 633. Medium-expressing clones showed intermediate but clearly above-baseline GFP levels (175, 46, and 89), while low-expressing clones were near baseline (29, 26, and 76). Overall, the expression ranking established by the MCB was largely preserved when assessed by flow cytometry, with the exception of clones exhibiting very low GFP expression. In these cases, the colony blot detected GFP signal more effectively than flow cytometry, likely due to spectral overlap between GFP emission in the FITC channel and chlorophyll autofluorescence. To further verify the correlation between GFP expression and the MCB results, we utilized an inverted fluorescent microscope to observe the GFP fluorescence in the high (H1, H2, H3), medium (M1, M2, M3), and low (L1, L2, L3) expressor clones. In red we can visualize chlorophyll's autofluorescence, and in green we can observe GFP's fluorescence (Fig 4). The fusion of the GFP to the Ble protein makes the fluorescence localize to the nucleus, due to the natural nuclear localization of the Ble protein [17]. In line with our previous findings, Fig 4 demonstrates that GFP fluorescence is significantly higher in the high expressor clones compared to the others, with medium expressors showing more fluorescence than the low expressor group.

**Microalgal colony blot applied to additional proteins and species**

To prove the broad applicability of the MCB to recombinant protein detection in algae, colony blots were performed on *C. reinhardtii* transformants expressing a range of heterologous proteins. As shown in Fig 5-A and Fig 5-B, multiple clones were screened for expression and secretion of a Flag® -tagged truncated version of the human protein ICAM-1.

The protein was expressed in the evolved strain *C. reinhardtii* CR25, and the colony blot was probed using an alkaline phosphatase conjugated anti-Flag® antibody as described in Torres-Tiji et al. 2022 [13]. Among the screened clones, one showed distinctly higher expression than all other clones, evidenced by a pronounced dark indigo halo surrounding the colony imprint. The isolated clone successfully showed high ICAM1 expression, verified by western blot and ligand LFA-1 binding, as previously reported [13].

Fig 5-C and Fig 5-D show the dot-blot screening used to isolate a *C. reinhardtii* clone expressing the SARS-CoV-2 spike protein receptor-binding domain (RBD) fused to cholera toxin subunit B (CtxB). As positive controls, clones previously shown to express the SARS-CoV-2 RBD fused to the mCloverGFP were included. In the second-to-last row, two control strains were spotted in duplicate: the first corresponds to a *C. reinhardtii* strain expressing an RBD-GFP fusion engineered for retention in the endoplasmic reticulum, and the second corresponds to a strain expressing and secreting the RBD fusion protein. The final row consists of triplicate spots of wild-type *C. reinhardtii* cells. The blot was incubated with a rabbit polyclonal anti-RBD primary antibody followed by a goat anti-rabbit alkaline phosphatase-conjugated secondary antibody. As shown, staining intensity was comparable across spots regardless of the size of the cell dot. Notably, the secreting RBD strain displayed weaker staining than the ER-retained control, likely reflecting lower overall recombinant protein accumulation. Several screened clones exhibited detectable RBD signal; however, only one clone showed staining intensity comparable to the positive controls. This clone was isolated, expanded in liquid culture, and analyzed by western blot. Cell lysates and culture supernatants were probed in parallel using rabbit polyclonal antibodies against RBD and CtxB, each detected with an alkaline phosphatase-conjugated anti-rabbit secondary antibody. Expression of the CtxB::RBD fusion protein was confirmed by the presence of a band at the expected molecular weight (~40 kDa) in both blots, which was absent in wild-type samples (S3 Fig).

Fig 5-E and Fig 5-F show an MCB in which *C. reinhardtii* and *C. pacifica* were transformed with the GFP-expressing plasmid described in the first Results section. The first square contains triplicate spots of wild-type *C. reinhardtii*. The second square contains triplicate spots of a GFP-*expressing C. reinhardtii* strain, while the third square contains triplicate spots of a GFP-expressing *C. pacifica* strain. The remaining squares contain *C. reinhardtii* clones exhibiting varying levels of GFP expression, as indicated by differences in staining intensity.

## Discussion

In summary, the MCB enables efficient screening of recombinant protein expression in *C. reinhardtii* and readily discriminates between high and low expressing clones. Western blot analysis confirmed the specificity of antibody binding and produced signal intensities that closely matched those seen in the MCB. As an independent validation step, flow cytometry analysis of cell populations derived from each isolated clone, the gold standard for quantifying fluorescent protein expression at the single-cell level, yielded results consistent with colony blot measurements. In addition, GFP fluorescence observed by inverted microscopy correlated with colony blot staining across the isolated clones. Together, these complementary validation approaches demonstrate that the MCB is a rapid, low-cost assay capable of assessing recombinant protein expression across hundreds of colonies simultaneously.

As shown in Figs 5-C and 5-E, staining intensity is highly reproducible across technical replicates of the same clone. Importantly, this reproducibility is observed despite unavoidable variability in colony growth, spotting, and final imprint size. In the MCB, cells are transferred onto the membrane as a densely packed biomass, forming a dot typically 1–2 mm in height. Once the membrane surface is saturated with cells, additional biomass deposition does not increase signal intensity, resulting in convergence of staining across replicate spots from the same strain. Consistent with this behavior, staining intensity does not correlate with the physical size of the colony imprint. Under these saturation conditions, signal intensity reflects recombinant protein abundance per cell rather than differences in total biomass or growth. This normalization by membrane saturation enables reliable identification and relative ranking of high-expressing clones and underpins the robustness of the MCB as a screening method.

The MCB has been validated using multiple alkaline phosphatase-conjugated antibodies targeting different antigens, all of which produced consistent results. Although the assay performs optimally when antibodies exhibit minimal nonspecific binding, moderate background signal can be tolerated provided that the signal from the protein of interest is clearly distinguishable from noise. An example of this is shown in Fig 5, panel D, where a clone expressing the CtxB:RBD fusion was successfully identified despite nonspecific binding of the anti-RBD antibody (S3 Fig). While the assay is not fully quantitative, it enables rapid identification of the highest-expressing clones from large populations. By screening hundreds of colonies, researchers can readily downselect the top 10 expressors for subsequent confirmation and precise quantification of recombinant protein expression using western blotting and ELISA, respectively, with the same antibody employed in the MCB assay.

Established high-throughput approaches such as FACS can screen thousands of cells per second, making clone generation rather than screening capacity the primary bottleneck. At this throughput, identification of high-expressing cells is straightforward, provided that high-level expression is biologically tolerated (e.g., not cytotoxic). In *C. reinhardtii*, however, random genomic integration and high rates of transgene truncation can lead to strong selection for fluorescent reporter expression decoupled from production of the intended fusion protein, particularly when extreme fluorescence phenotypes are selected. Additionally, in algal systems, intrinsic pigment autofluorescence can substantially interfere with detection of fluorescent reporters, reducing sensitivity and increasing false-positive rates when expression levels are low. Consequently, when strong expression is sought and fluorescent protein fusions do not compromise protein function or integrity, FACS remains the superior screening strategy.

While the MCB has been successfully applied in two algal species, *C. reinhardtii* and *C. pacifica*, our preliminary attempts to extend the approach to *Picochlorum celeri* were unsuccessful. These negative results may reflect low expression levels of the recombinant protein used rather than an inherent limitation of the method. Alternatively, they may be attributable to the comparatively robust cell wall of *P. celeri*, which could hinder efficient cell lysis. Because detection of intracellular recombinant proteins in this assay depends on cell disruption and transfer of cellular contents onto a nitrocellulose membrane, strains with more resilient cell walls may be less amenable to this approach. If this is the case, future studies may explore the applicability of the MCB in cell wall-deficient strains, which may further enhance assay sensitivity. Nevertheless, the simplicity of the technique and its minimal material requirements make it suitable for testing and optimization in additional algal species by researchers interested in evaluating its applicability in their own systems. Ultimately, the MCB provides a cost-effective, mid-throughput approach for detecting recombinant protein expression in *C. reinhardtii* and potentially other microalgal species. By circumventing the requirement for fluorescent reporters, which can interfere with the native folding or biological function of target proteins, the MCB allows for the screening of proteins in their intended form. By facilitating the rapid identification of high-performing clones for definitive downstream characterization, this method represents a practical and robust tool for accelerating the development of microalgal biotechnology.

## Supporting information

**S1 File: Step-by-step protocol, also available on protocols.io.**
(PDF)

**S2 Fig. Vector maps for all plasmids used in this study.** The first vector shows the test plasmid used to express mCloverGFP in Figures 1–4. The second vector shows the control plasmid used as the wild-type control in Figure 1. The third vector shows the plasmid used to express a secreted RBD::GFP fusion, used as a positive control for anti-RBD MCB in Figures 5C and 5D. The fourth vector shows an ER-retained RBD::GFP fusion, also used as a positive control in Figures 5C and 5D. The fifth vector shows the test plasmid used to express and secrete the CtxB::RBD fusion used in Figures 5E and 5F and Figure S3.
(TIF)

 

**S3 Fig. Anti-RBD and anti-CtxB western blot analysis of selected transformants shown in Figure 5D.** The left panel shows a western blot probed with an anti-CtxB antibody, and the right panel shows a western blot probed with an anti-RBD antibody. The first lane of each gel contains the protein ladder. Lanes 1–4 contain: (1) wild-type cell lysate, (2) wild-type culture supernatant, (3) clone 60 cell lysate, and (4) clone 60 culture supernatant.
(TIF)

## Acknowledgments

The authors would like to thank Professor Scott Biering at University of California San Diego for providing access to their inverted fluorescent microscope for cell imaging. The authors would like to thank Professors Susan and James Golden at University of California San Diego for providing access to their CytoFlex for flow cytometry analysis. The authors would also like to thank the US Department of Energy – Bioenergy Technologies Office for the continued funding support over the years.

## Author contributions

**Conceptualization:** Yasin Torres-Tiji, Francis J. Fields, Stephen P. Mayfield.

**Data curation:** Yasin Torres-Tiji.

**Funding acquisition:** Stephen P. Mayfield.

**Investigation:** Yasin Torres-Tiji.

**Methodology:** Yasin Torres-Tiji.

**Resources:** Francis J. Fields.

**Supervision:** Stephen P. Mayfield.

**Visualization:** Yasin Torres-Tiji.

**Writing – original draft:** Yasin Torres-Tiji.

**Writing – review & editing:** Yasin Torres-Tiji, Stephen P. Mayfield.

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
