## [Decision Letter · Decision Letter 0]

8 Sep 2025

Dear Dr. Torres-Tiji,

Thank you for submitting your manuscript to PLOS ONE. After careful consideration, we feel that it has merit but does not fully meet PLOS ONE’s publication criteria as it currently stands. Therefore, we invite you to submit a revised version of the manuscript that addresses the points raised during the review process.

We look forward to receiving your revised manuscript.

Kind regards,

Rishiram Ramanan

Academic Editor

PLOS ONE

Journal Requirements:

2. Thank you for stating the following financial disclosure: [US Department of Energy - Bioenergy Technologies Office, grant number: DE-EE0009671 (APEX)].

3. Thank you for stating the following in the Competing Interests section: [The authors disclose the following potential competing interests: Stephen Mayfield received funding from the U.S. Department of Energy and holds board membership as well as equity in Algenesis Inc.].

We note that one or more of the authors are employed by a commercial company: [Algenesis Inc.]

4. We noted in your submission details that a portion of your manuscript may have been presented or published elsewhere. [Yes, Figure 1 is also published in the linked Protocols.io protocol. It is not a dual publication because both this manuscript and the protocols.io are linked as part of the same publication.] Please clarify whether this publication was peer-reviewed and formally published. If this work was previously peer-reviewed and published, in the cover letter please provide the reason that this work does not constitute dual publication and should be included in the current manuscript.

5. We note that your Data Availability Statement is currently as follows: [All relevant data are within the manuscript and its Supporting Information files.]

Reviewers' comments:

Reviewer's Responses to Questions

**Comments to the Author**



Reviewer #1: Yes

Reviewer #2: Yes

2. Has the protocol been described in sufficient detail?

To answer this question, please click the link to protocols.io in the Materials and Methods section of the manuscript (if a link has been provided) or consult the step-by-step protocol in the Supporting Information files.

Reviewer #1: Partly

Reviewer #2: Partly

3. Does the protocol describe a validated method?

Reviewer #1: Yes

Reviewer #2: Yes

4. If the manuscript contains new data, have the authors made this data fully available?

Reviewer #1: N/A

Reviewer #2: N/A

**5. Is the article presented in an intelligible fashion and written in standard English?**

Reviewer #1: Yes

Reviewer #2: Yes

Reviewer #1: The manuscript presents an innovative and accessible protocol for identifying Chlamydomonas reinhardtii transformants with high recombinant protein expression directly on agar plates using colony blotting and colorimetric Western detection. The approach is simple, low-cost, and potentially adaptable to other microalgae. However, the current version would benefit from additional methodological details, quantitative validation, and broader applicability testing. While some issues can be addressed by minor editorial revisions, several aspects require additional experimentation or data analysis before publication.

Major Comments:

The manuscript describes the method as “semi-quantitative,” yet it lacks a standardized metric for ranking expression levels. Incorporating a protocol for image-based densitometry, such as using ImageJ to measure pixel intensity with calibration against known protein quantities, would address this limitation.

Although the method is presented as broadly applicable to other microalgae, it has only been validated in C. reinhardtii. Expanding validation to include at least one additional protein beyond GFP, and ideally another algal species such as Nannochloropsis or Phaeodactylum tricornutum, would strengthen the generalizability of the approach.

Furthermore, the manuscript does not provide any benchmarking against established screening techniques like FACS or ELISA; a comparative analysis of throughput, sensitivity, and false positive rates would provide valuable context. The method also assumes consistent biomass transfer per spot, but variability in growth or pipetting could influence signal intensity.

In addition, correlation between colony halo intensity and actual protein levels is shown for only a few strains (H13, M13, L1–L3); presenting a broader correlation analysis with R² values for all tested strains would strengthen the dataset. The observation that intracellular proteins can be detected via passive release from cell death is intriguing; however, supporting data confirming whether expression ranking on plates corresponds to trends in liquid culture would be useful.

Minor Comments:

While the vector is described in the text, an annotated plasmid map would aid comprehension.

The authors should also clarify whether PVDF membranes were tested and, if so, compare their performance with nitrocellulose.

The section spanning lines 174–193 contains a mix of methods and results, which should be reorganized for clarity.

Quantitative fluorescence measurements from a plate reader could complement qualitative microscopy observations.

Figure 3 would benefit from the addition of a scale bar to enhance interpretability.

Reviewer #2: Amazing work. I have attempted a similar approach for large scale screening of nuclear expressed Strep-II tagged proteins however smudging of the blot and unspecific binding was always made it difficult to identify high expressing transformants. Your combination of lysis, flag tag and development technique used clearly overcomes these issues. It has always been surprising this was not a standardized technique for the microalgae community.

Some constructive criticism I could provide would be to attempt this with a cell wall deficient strain (eg. CC-4350) also commonly used for transformation to see if the blotting is still well defined, although possibly not needed for this manuscript.

I would also appreciate the specification of the Alkaline-Phosphatase conjugated antibody used (e.g. manufacturer) as I can't seem to find reference to in their the main text or in the protocols.io version. This would be useful for others to replicate the is work.

Greatly appreciate that this was shared via protocols.io. This protocol will be of great use to the community in cases where fluorescently tagged recombinant proteins are not appropriate. I look forward to attempting this in the near future.

**Do you want your identity to be public for this peer review?** For information about this choice, including consent withdrawal, please see our Privacy Policy

Reviewer #1: No

Reviewer #2:**Yes**

---

## [Author Response · Author response to Decision Letter 1]

31 Jan 2026

Please see the uploaded “Response to Reviewers” document for a detailed, point-by-point response to all reviewer and editor comments.

---

## [Decision Letter · Decision Letter 1]

6 Mar 2026

Microalgal Colony Blot: a simple and rapid method for direct detection of recombinant protein production in microalgae colonies

PONE-D-25-38531R1

Dear Dr. Torres-Tiji,

We’re pleased to inform you that your manuscript has been judged scientifically suitable for publication and will be formally accepted for publication once it meets all outstanding technical requirements.

Kind regards,

Rishiram Ramanan

Academic Editor

PLOS One

Additional Editor Comments (optional):

Reviewers' comments:

Reviewer's Responses to Questions

**Comments to the Author**



Reviewer #2: Yes

2. Has the protocol been described in sufficient detail?

To answer this question, please click the link to protocols.io in the Materials and Methods section of the manuscript (if a link has been provided) or consult the step-by-step protocol in the Supporting Information files.

Reviewer #2: Yes

3. Does the protocol describe a validated method?

Reviewer #2: Yes

4. If the manuscript contains new data, have the authors made this data fully available?

Reviewer #2: Yes

**5. Is the article presented in an intelligible fashion and written in standard English?**

Reviewer #2: Yes

Reviewer #2: The current revisions have improved an already acceptable manuscript and added additional details that would allow others to more easily use this protocol. I appreciate the trial with related Chlamydomonas, as well as more distantly related species. The writing is clear and concise. I look forward to citing this paper in the near future.

One slight error, in line 183, there is reference to a supp figure SX - I assume this would be S2.

**Do you want your identity to be public for this peer review?** For information about this choice, including consent withdrawal, please see our Privacy Policy

Reviewer #2: No

---

## [Editor Report · Acceptance letter]

PONE-D-25-38531R1

PLOS One

Dear Dr. Torres-Tiji,

I'm pleased to inform you that your manuscript has been deemed suitable for publication in PLOS One. Congratulations! Your manuscript is now being handed over to our production team.

Kind regards,

on behalf of

Dr. Rishiram Ramanan

Academic Editor

PLOS One